# Towards Plastic and Stable Exemplar-Free Incremental Learning: A Dual-Learner Framework with Cumulative Parameter Averaging

## Abstract

The dilemma between plasticity and stability presents a significant challenge in Incremental Learning (IL), especially in the exemplar-free scenario where accessing old-task samples is strictly prohibited during the learning of a new task. A straightforward solution to this issue is learning and storing an independent model for each task, known as Single Task Learning (STL). Despite the linear growth in model storage with the number of tasks in STL, we empirically discover that averaging these model parameters can potentially preserve knowledge across all tasks. Inspired by this observation, we propose a **D**ual-**L**earner framework with **C**umulative **P**arameter **A**veraging (DLCPA). DLCPA employs a dual-learner design: a plastic learner focused on acquiring new-task knowledge and a stable learner responsible for accumulating all learned knowledge. The knowledge from the plastic learner is transferred to the stable learner via cumulative parameter averaging. Additionally, several task-specific classifiers work in cooperation with the stable learner to yield the final prediction. Specifically, when learning a new task, these modules are updated in a cyclic manner: i) the plastic learner is initially optimized using a self-supervised loss besides the supervised loss to enhance the feature extraction robustness; ii) the stable learner is then updated with respect to the plastic learner in a cumulative parameter averaging manner to maintain its task-wise generalization; iii) the task-specific classifier is accordingly optimized to align with the stable learner. Experimental results on CIFAR-100 and Tiny-ImageNet show that DLCPA outperforms several state-of-the-art exemplar-free baselines in both Task-IL and Class-IL settings[1].

## 1 Introduction

Incremental learning (IL) refers to the ability to continuously learn new knowledge from a series of tasks, which is crucial for many open-world applications such as autonomous robots. It has thus garnered significant attention in recent years (Mai et al., 2022; Delange et al., 2021). Typically, a model is expected to sequentially learn from a series of tasks, with samples from a completed task becoming inaccessible for future learning (Van de Ven & Tolias, 2019). In such a context, an effective IL learner is expected to exhibit both high **plasticity** for new task learning and **stability** to retain old-task knowledge. However, few established IL methods can achieve a perfect balance between them, known as the plasticity-stability dilemma (Mermillod et al., 2013).

Existing IL methods fall into three categories: memory-based, regularization-based, and parameter-isolation-based. Memory-based methods (Buzzega et al., 2020; Arani et al., 2022) maintain an additional memory for storing old-task exemplars and use this stored information to recall old-task knowledge when learning a new task. While they exhibit high plasticity for new knowledge learning, old-task knowledge may be overwritten, especially when stored exemplars cannot support the old-task data distribution. Regularization-based methods (Kirkpatrick et al., 2017; Zhu et al., 2021) penalize changes in essential neurons or activation through regularization terms, typically setting a

---

[1]The source code is available in supplementary materials.

hyper-parameter to balance plasticity and stability. However, balancing this weight in real scenarios is challenging due to the randomness and disorder of deep neural network training. Parameter-isolation-based methods (Hung et al., 2019; Sun et al., 2023b) allocate network parameters (or parameter basis) to specific tasks, preventing them from being updated in subsequent task learning. While they often maintain stability, plasticity gradually decreases as network capacity is consumed.

Instead of following the established techniques, this paper turns to investigate single task learning (STL), an upper-bound IL with both high plasticity and stability. As depicted in Figure 1 (a), STL trains a specific model for each new task, leaving trained models unchanged. However, memory utilization increases linearly with the number of learning tasks, limiting the practicability of STL. To overcome this limitation, one intuitive solution is to consolidate the knowledge from all STL models into a single unified learner. In the context of exemplar-free IL, it usually necessitates an intricate design to apply advanced distilling techniques during the learning process (Zhu et al., 2021). Instead, we explore a simple strategy, which involves averaging all STL feature extractors in the parameter space, as shown in Figure 1 (b). This strategy showcases considerable potential in retaining knowledge across all tasks. Furthermore, with suitable finetuning of classifiers, it can approximate the upper-bound performance achieved by STL.

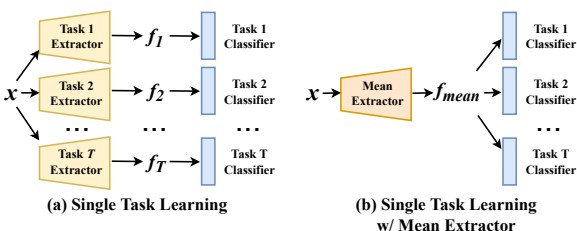

Figure 1: Diagram of Single Task Learning (STL) and its variation. (a) STL represents an upper-bound IL model that trains a specific network for each task. (b) STL-me is an STL variation that averages all STL feature extractors in the parameter space.

In light of the aforementioned observations, we propose a **D**ual-**L**earner framework with **C**umulative **P**arameter **A**veraging (DLCPA) for exemplar-free IL. DLCPA is characterized by a dual-learner design: a plastic learner for acquiring new task knowledge, and a stable learner for accumulating all previously learned knowledge. Additionally, several task-specific classifiers cooperate with the stable learner to achieve the final prediction. During each task training, these three modules are updated alternately. The plastic learner initially adapts to new task data using a self-supervised loss in conjunction with a supervised loss to enhance feature extraction robustness. Subsequently, the stable learner is updated in relation to the plastic learner using a cumulative average approach to integrate new task feature-extraction knowledge. Finally, the task-specific classifier is optimized to align with the features extracted by the stable learner. These three steps form a cohesive loop that is executed until convergence.

The contributions of this work are summarized as follows:

- Our empirical findings suggest that averaging the STL models in the parameter space can potentially consolidate knowledge across all tasks. In conjunction with finetuned task-specific classifiers, the averaged model can approximate the performance of STL, an upper bound of exemplar-free IL.

- Inspired by the above observation, we propose the DLCPA framework for exemplar-free IL. DLCPA disentangles the dilemma between plasticity and stability via a dual-learner scheme. A cumulative parameter averaging strategy is introduced to facilitate a gentle knowledge transfer between the dual learners. Moreover, meticulously designed updating rules ensure the overall performance of the framework.

- We evaluate DLCPA on CIFAR-100 and Tiny-ImageNet. The experimental results show that DLCPA outperforms several state-of-the-art baselines under both exemplar-free Task-IL and Class-IL settings.

## 2 RELATED WORK

Incremental learning has been widely studied in recent years (Mai et al., 2022; Zhou et al., 2023), and many methods have emerged. According to the techniques employed for preventing catastrophic

forgetting (McCloskey & Cohen, 1989), they can be categorized into three classes: memory-based methods, regularization-based methods, and parameter-isolation-based methods.

Most memory-based methods rely on exemplar memory (Chaudhry et al., 2021; Cha et al., 2021; Kang et al., 2022; Bonicelli et al., 2022), replaying old-task exemplars with new data to retain old-task knowledge. Some memory-based methods (Shin et al., 2017; Yin et al., 2020) use sample generation techniques for IL in the exemplar-free scenario, replaying models with generated pseudo old-task samples or features. Regularization-based methods, typically designed for exemplar-free Task-IL, use knowledge distillation to maintain network activation during learning (Li & Hoiem, 2017; Douillard et al., 2020), or penalize changes in essential neurons during IL (Zenke et al., 2017; Kirkpatrick et al., 2017; Aljundi et al., 2018). Parameter-isolation-based methods allocate parameters for specific tasks. Some mainstream technology branches include the expansion-based methods (Hu et al., 2021; Rusu et al., 2016) assigning network branches for each task, the mask-based methods (Mallya & Lazebnik, 2018; Mallya et al., 2018; Ke et al., 2020) preserving crucial parameters for each task, and projection-based methods (Saha et al., 2021; Wang et al., 2021) constraining the gradient descent direction to prevent interference with model performance for old tasks.

This paper introduces DLCPA, a novel exemplar-free incremental learning framework with a dual-learner architecture. Similar architectures are found in exemplar-based IL methods like CLS-ER (Arani et al., 2022) and DualNet (Pham et al., 2021), which use a plastic model for new knowledge and a stable model for long-term knowledge. However, these methods depend on old-task exemplars to maintain the stable model and prevent forgetting. In contrast, DLCPA employs a cumulative average update strategy for the stable learner, achieving a desirable balance between plasticity and stability in an exemplar-free setting.

## 3 ANALYSIS

The goal of this section is to present our empirical findings that averaging model in the parameter space of STL could potentially be an effective strategy for preserving knowledge across all tasks. These findings serve as the motivation for our DLCPA, which will be introduced in Section 4.

### 3.1 PROBLEM DEFINITION

In IL, a model is tasked with learning knowledge from a sequence of $T$ task datasets: $D_1, D_2, \ldots, D_T$. This paper specifically focuses on IL under the exemplar-free setting(Masana et al., 2022), i.e., accessing or storing old-task samples is strictly forbidden during learning a new task. In particular, when training task $t$, only the corresponding dataset $D_t = \{(x_k^t, y_k^t)\}_{k=1}^{n_t}$ is accessible, where $(x_k^t, y_k^t)$ represents an input-output pair and $n^t$ denotes the number of samples in $D_t$. Furthermore, the categories in different tasks are disjoint, i.e., $Y_{t_1} \cap Y_{t_2} = \emptyset$ for $t_1 \neq t_2$, with $Y_t$ representing the label set of task $t$.

The objective of IL is to encapsulate all task knowledge within a single model. During testing, we assess model performance using all $T$ task samples. Task identifiers are provided alongside query samples in Task-IL, but not in Class-IL. The DLCPA proposed in this paper is applicable to both Task-IL and Class-IL scenarios.

### 3.2 BASELINE: SINGLE TASK LEARNING

Single Task Learning (STL) is recognized as an upper-bound baseline for exemplar-free IL(Yoon et al., 2020; Saha et al., 2021). As depicted in Figure 1 (a), STL trains an independent network for each task[2]. Given a query sample $x$ from task $t$, STL employs the model corresponding to task $t$ to make a prediction, which can be formulated as:

$$\hat{y}_{stl} = \Gamma_t^\top f(x \mid \Theta_t), \tag{1}$$

where $f(\cdot)$ represents the feature extractor (the model excluding the last layer of the network) and $\Theta_t$ denotes its parameters. Additionally, $\Gamma_t$ signifies the parameter matrix of the classifier (the final layer of the model) corresponding to task $t$, with each column of $\Gamma_t$ storing a class prototype.

---

[2]This work explores a stationary version of STL that initializes the new-task model $\Theta_t$ with the preceding $\Theta_{t-1}$. Please refer to Algorithm 2 in the Appendix for detailed pseudo codes.

STL boasts high plasticity by introducing an independent model for each task, thereby facilitating unrestricted new knowledge learning. Concurrently, the old-task models of STL remain unchanged, ensuring high stability. However, the memory usage of STL escalates linearly with the number of learned tasks, restricting its practical application. To address this limitation, a straightforward idea is to consolidate the knowledge from all STL models into a unified learner, aiming to approximate the performance of STL. Under the exemplar-free IL setting, it usually necessitates an intricate design to apply advanced distilling techniques during the learning process (Zhu et al., 2021). Instead, we investigate a simple strategy that averages all feature extractors in the parameter space, which showcases significant potential in preserving the knowledge across all tasks, thereby approximating the performance of STL. We elaborate on our findings in the following subsection.

Table 1: Accuracy (%) under Task-IL of STL and its variations on MNIST and CIFAR-100.

| Methods | 5-split MNIST | 10-split CIFAR-100 |
|---|---|---|
| STL | 99.73±0.1 | 86.94±0.3 |
| STL-me | 96.74±0.7 | 59.71±1.0 |
| STL-me w/ classifier finetuning | 98.55±0.3 | 80.65±0.2 |

### 3.3 STL APPROXIMATION WITH MEAN EXTRACTOR

To address the storage consumption issue of STL, we explore a simple solution, STL-me, which averages all feature extractors into a unified one and makes predictions based on this mean feature extractor, as shown in Figure 1 (b). Specifically, given a query sample $x$ from task $t$, STL-me makes a prediction as follows:

$$\hat{y}_{stl\_me} = \Gamma_t^\top f\left(x \mid \frac{1}{T} \sum_{i=1}^T \Theta_i\right). \tag{2}$$

To evaluate the performance of STL-me, we conduct experiments on two datasets: MNIST and CIFAR-100. As depicted in Table 1, the performance of this straightforward approach fails to meet our expectations. There exists an evident accuracy gap between STL and STL-me, particularly on CIFAR-100. A possible reason for this performance degradation could be that averaging extractor parameters disrupts the sample distribution of each category in the feature space, thereby undermining the effectiveness of class prototypes in each task-specific classifier. To address this issue, we finetune each task-specific classifier to adapt to the mean feature extractor using the data of the corresponding task. The prediction is then made based on the finetuned classifier parameters $\Gamma_t^\star$:

$$\hat{y}_{stl\_me\_ft} = \Gamma_t^{\star\top} f\left(x \mid \frac{1}{T} \sum_{i=1}^T \Theta_i\right). \tag{3}$$

The last row Table 1 displays that STL-me with classifier finetuning achieves quite competitive performance[3]: it only trails STL by 1% and 6%[4] in accuracy on MNIST (LeCun & Cortes, 2010) and CIFAR-100 (Krizhevsky, 2012), respectively. These results underscore the effectiveness of the mean feature extractor, which shows promise in consolidating knowledge across all tasks. Moreover, the average operation significantly reduces the storage consumption from $O(T)$ to $O(1)$. While finetuning task-specific classifiers based on the average feature extractor is an unlikely task in exemplar-free IL, it still shows significant potential to inspire a promising IL method. Motivated by these observations, we propose our DLCPA framework, which will be detailed in the next section.

## 4 METHODS

### 4.1 OVERVIEW

The proposed DLCPA framework is characterized by three main components: a plastic leaner, a stable learner, and $T$ task-specific classifiers.

---

[3]See Section C in the Appendix for more detailed evidence.

[4]Current IL methods are still far from STL (Saha et al., 2021), and a 6% accuracy loss on CIFAR-100 in this situation is acceptable

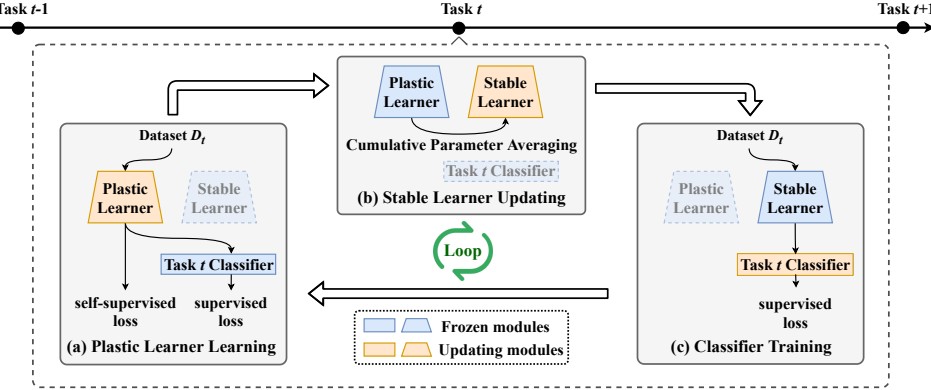

Figure 2: Diagram of task $t$ training processes of DLCPA.

- **Plastic learner** aims to rapidly acquire knowledge from a new task. It is implemented as a deep neural network with ReLU activation units. Given an input $x$, it yields a feature representation $f(x \mid \Theta)$, where $\Theta$ denotes the network parameters.
- **Stable learner** is responsible to accumulate and retain the knowledge learned by the plastic learner over time. It shares the same architecture as the plastic learner but maintains its own parameters, denoted as $\Phi$. Hence, the stable learner can also extract a feature $f(x \mid \Phi)$ for any given input $x$.
- **Classifier** is a linear layer to perform classification based on the features extracted by the plastic learner or the stable learner. Each task is associated with a specific classifier. We denote the parameter of the classifier for task $t$ as $\Gamma_t$.

As depicted in Figure 2, the training of the DLCPA framework for task $t$ involves a three-stage loop: plastic learner learning, stable learner updating, and classifier training. In the plastic learner learning stage, we employ self-supervised techniques to guide the model in learning robust parameters. In the stable learner updating stage, an updating strategy with cumulative parameter averaging is designed to transfer the acquired knowledge from the plastic learner to the stable learner. In the classifier training stage, the task-specific classifier is trained to fit the supervision knowledge based on the features extracted by the stable learner. These three stages are iteratively performed several times.

During the inference phase, given a query sample $x$, DLCPA first utilizes the stable learner to extract its feature, which is then fed into the classifier to yield the prediction. For task-IL, we select the classifier corresponding to the task label (for instance, $t$) of a query:

$$\hat{y} = \Gamma_t^\top f(x \mid \Phi), \tag{4}$$

where $\hat{y}$ represents the model prediction. For Class-IL, we concatenate the parameters of all task classifiers into a unified one, denoted as $\Gamma = [\Gamma_1, \ldots, \Gamma_T]$, to make a prediction:

$$\hat{y} = \Gamma^\top f(x \mid \Phi). \tag{5}$$

Next, we elaborate on the three learning stages in the following three subsections.

### 4.2 PLASTIC LEARNER LEARNING

During the plastic learner learning stage, as shown in Figure 2 (a), both the stable learner and the classifier are frozen, and only the plastic learner is trained to learn new-task representation. Particularly, we employ self-supervised learning techniques to guide the plastic learner's fitting process. This approach is motivated by two key factors. On the one hand, previous studies (Wu et al., 2021; Fini et al., 2022) have shown that self-supervised technique can implicitly guide a neural model to learn task-independent knowledge for incremental learning. It, therefore, can encourage the plastic learner to learn unbiased parameters that are more suitable for the cumulative parameter averaging step[5]; On the other hand, self-supervised learning tends to discover transferable knowledge (Jing

---

[5]See Figure 5 in the Appendix for illustration.

& Tian, 2021), which boosts DLCPA to forward even backward transfer knowledge. In addition, a supervised loss is utilized to enable the model to learn discriminative representations for new tasks.

Technically, we wrap the plastic learner with a self-supervised module and apply the self-supervised loss along with a supervised loss to optimize the plastic learner parameter $\Theta_t$ for task $t$:

$$\min_{\Theta_t} \mathbb{E}_{(x,y)\sim D_t} L_{ssl}(f(x \mid \Theta_t)) + \lambda L_{sl}(\Gamma_t^\top f(x \mid \Theta_t), y), \tag{6}$$

where $L_{ssl}(.)$ and $L_{sl}(.,.)$ denote a self-supervised loss and the cross-entropy supervised loss, respectively. $\lambda$ is a hyper-parameter that balances the two losses. The choice of self-supervised learners is not restricted, and we provide results with BYOL (Grill et al., 2020), SimClr (Chen et al., 2020), and MoCoV2 (He et al., 2020) in the experiments.

### 4.3 STABLE LEARNER UPDATING

Upon updating, the plastic learner acquires new-task knowledge, enhancing its ability to extract informative features for the new task. On the other hand, the stable learner prioritizes generalizable knowledge across different tasks. To facilitate a gentle transfer of knowledge from the plastic learner to the stable learner without compromising its stability, we introduce a cumulative average updating strategy, which is detailed below.

As observed in Section 3.3, the mean feature extractor can potentially encapsulate knowledge across all tasks. This observation motivates us to construct a stable learner by averaging the parameters of the plastic learner across all tasks:

$$\Phi_t = \frac{\sum_{i=1}^{t} \Theta_i}{t}, \tag{7}$$

where $\Phi_t$ denotes the stable learner parameters after learning task $t$.

However, equation 7 necessitates storing the plastic learner parameters for all tasks, leading to memory usage that grows linearly with the number of learned tasks. To address this, we propose a cumulative average updating strategy for the stable learner parameters, achieving the same goal as Eq. equation 7 but with fixed memory usage. Specifically, before learning task $t$, we store a copy of the stable learner, whose parameters can be viewed as the average of the plastic learner parameters for all previous tasks, i.e., $\Phi_{t-1} = \frac{\sum_{i=1}^{t-1} \Theta_i}{t-1}$. Consequently, we can achieve equation 7 as follows:

$$\Phi_t = \frac{\Theta_t + (t-1) * \Phi_{t-1}}{t}. \tag{8}$$

Equation 8 only requires storing the old-task extractor $\Phi_{t-1}$, making the updating memory-efficient.

### 4.4 CLASSIFIER TRAINING

Next, as depicted in Figure 2 (c), DLCPA updates the classifier based on the features extracted by the stable learner, aiming to align the classifier with the stable learner. We halt the gradient backpropagation beyond the classifier to prevent the stable learner from forgetting. The loss function for training the classifier is defined as:

$$\min_{\Gamma_t} \mathbb{E}_{(x,y)\sim D_t} L_{sl}(\Gamma_t^\top f(x \mid \Phi_t), y). \tag{9}$$

Upon completion of classifier training, one iteration for task $t$ finishes. Notably, within each task, the three-step iteration **repeats** multiple times until the model converges. We outline the training process of DLCPA in Algorithm 1 in the Appendix.

### 4.5 FURTHER DISCUSSION

In each iteration, although the plastic learner and the classifier are updated separately, they actually interact through the stable learner. On the one hand, the classifier is trained to align with the new-task feature representations yielded by the stable learner, which has encompassed the knowledge acquired by the plastic learner through accumulative parameter averaging. On the other hand, the

training of the plastic learner is guided by the class prototypes within the classifier. These prototypes encapsulate the task-general information learned by the stable learner, thereby encouraging the plastic learner to capture more generalizable patterns. DLCPA benefits from this indirect communication, which is empirically substantiated by ablation studies presented in Table 4.

## 5 EXPERIMENTS

### 5.1 SETTINGS

**Datasets.** We select the following two widely used datasets: CIFAR-100 (Krizhevsky, 2012) and Tiny-ImageNet (Stanford, 2015). CIFAR-100 has 100 classes containing 600 images each. The Tiny-ImageNet dataset contains 200 classes, and 600 images for each class. In our experiments, they are **equally** separated into 10, 20, and 25 incremental branches.

**Evaluation metrics.** Following Buzzega et al. (2020), we take the classification accuracy (ACC %) after learning all tasks as the primary metric for the performance evaluation. Besides, we also report backward knowledge transfer (BWT %) (Lopez-Paz & Ranzato, 2017), which formulated as:

$$BWT = \frac{1}{T-1} \sum_{i=1}^{T-1} R_{T,i} - R_{i,i}, \tag{10}$$

where $R_{i,j}$ denotes the test classification accuracy of the model on task $j$ after learning task $i$. Furthermore, to alleviate the influence of randomness in neural network training, we run all experiments five times with random seeds and report the average performance.

**Baselines:** We compare our method with various latest and classic incremental learning methods, including Episodic Memory (GEM) (Lopez-Paz & Ranzato, 2017), Dark Experience Replay (DER and DER++) (Buzzega et al., 2020), Learning without Forgetting (LwF) (Li & Hoiem, 2017), Elastic Weight Consolidation (EWC) (Kirkpatrick et al., 2017), PackNet (Mallya & Lazebnik, 2018), Orthogonal Weights Modification (OWM) (Zeng et al., 2019), MUC-MAS (Liu et al., 2020), PASS (Zhu et al., 2021), Adam-NSCL (Wang et al., 2021), Gradient Projection Memory (GPM) (Saha et al., 2021), Bit-Level Information Preserving (BLIP) (Shi et al., 2021), ADNS (Kong et al., 2022), VDFD (Li et al., 2023), ILCOC (Sun et al., 2021), Always Be Dreaming (ABD) (Smith et al., 2021), Filter Atom Swapping (FAS) (Miao et al., 2022), and DCPOC (Sun et al., 2023a).

**Implementation details.** We employ the ResNet18 (He et al., 2016) as the backbone architecture. All hyperparameters are searched through a validation set which is constructed by sampling ten percent samples from the training set. For DLCPA, the learning rate is set to 0.005 and 0.02, and the batch size is set to 32 and 512 for CIFAR-100 and Tiny-ImageNet, respectively. The network parameters are optimized by SGD for 100 epochs per task. The balance weight $\lambda$ is set to 10 for both datasets. More details can be found in our supplementary code. Besides, we provide the performance of DLCPA with several self-supervised learners, including BYOL (Grill et al., 2020), SimClr (Chen et al., 2020), and MoCoV2 (He et al., 2020). Notably, we refresh the negative feature queue of MoCoV2 at the beginning of each task to satisfy the exemplar-free setting.

### 5.2 PERFORMANCE COMPARISON

We first follow Li et al. (2023) to compare DLCPA with several baselines under the Task-IL setting. Table 2 reports the classification accuracy and BWT on 10-split CIFAR-100, 20-split CIFAR-100, and 25-split Tiny-ImageNet. On CIFAR-100, DLCPA equipped with BYOL achieves the second-best performance in the 10-split setting, trailing VDFD only by 0.15%. Additionally, DLCPA performs the best in the 20-split CIFAR-100, with the BYOL version surpassing the second-best VDFD by 0.84%. In the case of 25-split Tiny-ImageNet, DLCPA outperforms all existing methods, exceeding the second-best GPM by 9.27%. Moreover, DLCPA shows positive BWT in 25-split Tiny-ImageNet because the introduced self-supervised techniques encourage DLCPA to learn task-independent knowledge, which benefits DLCPA's old-task performance.

We next follow Buzzega et al. (2020) to evaluate the Class-IL performance of DLCPA, and report the experimental results on 10-split CIFAR-100 and 10-split Tiny-ImageNet in Table 3. Our findings show that the Class-IL performance of DLCPA is highly competitive. Compared with the second-best performance, DLCPA is 2.44% higher on CIFAR-100 and 0.73% higher on Tiny-ImageNet.

Table 2: Task incremental learning results on 10-split CIFAR-100, 20-split CIFAR-100, and 25-split Tiny-ImageNet. (*) indicates the upper-bound model that trains a specific model for each task. (-) means that the result was unavailable, due to the intractable training time by our implementation (GEM). **Bold** and underlined denote the best and the second-best performance.

| Methods | Buffer | 10-split CIFAR-100 ACC (%) | 10-split CIFAR-100 BWT (%) | 20-split CIFAR-100 ACC (%) | 20-split CIFAR-100 BWT (%) | 25-split Tiny-ImageNet ACC (%) | 25-split Tiny-ImageNet BWT (%) |
|---|---|---|---|---|---|---|---|
| STL* | - | 86.94 | 0.00 | 87.79 | 0.00 | 80.30 | 0.00 |
| GEM | 500 | 61.59 | -26.40 | 71.25 | -9.22 | - | - |
| DER | 500 | 73.26 | -13.69 | 77.37 | -5.16 | 57.12 | -28.49 |
| DER++ | 500 | 74.86 | -12.56 | 77.82 | -3.57 | 55.09 | -28.98 |
| LWF | - | 70.70 | -6.27 | 74.38 | -9.11 | 56.57 | -11.19 |
| EWC | - | 71.28 | -2.97 | 70.90 | -3.03 | 52.33 | -6.17 |
| PackNet | - | 77.18 | **0.00** | 67.50 | 0.00 | 52.93 | 0.00 |
| OWM | - | 68.89 | -1.88 | 68.47 | -3.37 | 49.98 | -3.64 |
| MUC-MAS | - | 63.73 | -3.88 | 67.22 | -5.72 | 41.18 | -4.04 |
| PASS | - | 71.23 | -5.21 | 74.43 | -4.03 | 56.78 | -5.33 |
| Adam-NSCL | - | 73.77 | -1.60 | 75.95 | -3.66 | 58.28 | -6.05 |
| GPM | - | 70.93 | -3.52 | 77.55 | -1.20 | 69.48 | -5.29 |
| BLIP | - | 61.09 | -0.70 | 68.17 | -4.21 | 47.63 | -5.79 |
| ADNS | - | 77.21 | -2.32 | 77.33 | -3.25 | 59.77 | -4.58 |
| VDFD | - | **83.30** | -1.27 | 85.84 | -1.53 | 65.99 | -2.35 |
| DLCPA + BYOL | - | 83.15 | -0.04 | **86.68** | 0.27 | 76.90 | 0.56 |
| DLCPA + SimClr | - | 82.03 | -0.99 | 85.95 | -0.08 | 77.71 | **3.94** |
| DLCPA + MoCoV2 | - | 81.15 | -0.34 | 85.56 | **0.31** | **78.75** | 3.91 |

Table 3: Class incremental learning results (ACC %) on 10-split CIFAR-100 and 10-split Tiny-ImageNet. (*) indicates the upper-bound model that is jointly trained with all tasks. (†) imply the results are quoted from VDFD, in which the standard deviations are not provided.

| Methods | Buffer | 10-split CIFAR-100 | 10-split Tiny-ImageNet |
|---|---|---|---|
| Joint* | - | 70.31 | 58.07 |
| GEM | 500 | 11.76±0.7 | - |
| DER | 500 | 34.24±1.4 | 17.75±1.1 |
| DER++ | 500 | 36.52±2.0 | 19.38±1.4 |
| OWM | - | 27.63±0.5 | 15.30±0.3 |
| ILCOC | - | 22.19±1.6 | 15.78±0.4 |
| PASS | - | 31.80±0.7 | 28.48±0.6 |
| ABD | - | 33.30±0.3 | 15.80±0.4 |
| FAS | - | 25.79±0.7 | 24.29±0.3 |
| DCPOC | - | 25.80±0.4 | 19.75±0.1 |
| VDFD† | - | 38.38 | 26.21 |
| DLCPA + BYOL (ours) | - | 40.67±1.1 | 23.19±0.6 |
| DLCPA + SimClr (ours) | - | **40.82±0.4** | 26.68±0.3 |
| DLCPA + MoCoV2 (ours) | - | 40.11±0.8 | **29.21±0.3** |

The competitive Class-IL performance of DLCPA is attributed to the task-wise balanced feature extractor, which is the average of each task's optimal extractor and does not suffer the notorious bias problem in Class-IL (Wu et al., 2019). We empirically prove that DLCPA is task-wise balanced via the confusion matrix in Section E.2 of the Appendix.

Table 4: Ablation experiment results (ACC %) of DLCPA on CIFAR-100 and Tiny-ImageNet.

| Methods | 10-split CIFAR-100 Class-IL | 10-split CIFAR-100 Task-IL | 10-split Tiny-ImageNet Class-IL | 10-split Tiny-ImageNet Task-IL |
|---|---|---|---|---|
| (a) Plastic learner learning w/o SSL | 27.93±1.5 | 80.90±1.0 | 18.62±0.4 | 57.46±0.4 |
| (b) Plastic learner learning w/ additional classifier | 35.53±0.6 | 78.35±0.5 | 25.90±0.4 | 62.64±0.4 |
| (c) Stable learner updating w/ EMA | 30.00±1.8 | 74.87±0.6 | 24.25±0.3 | 61.81±0.2 |
| (d) Stable learner updating w/ directly copying | 22.98±1.3 | 72.56±1.3 | 20.92±0.3 | 58.59±0.5 |
| (e) Classifier training w/ plastic-learner feature | 13.67±1.3 | 64.73±2.3 | 22.23±0.2 | 60.34±0.2 |
| DLCPA + MoCoV2 (ours) | **40.11±0.8** | **81.15±0.5** | **29.21±0.3** | **65.90±0.4** |

## 5.3 ABLATION STUDY

**Effectiveness of plastic learner learning stage.** We first construct two variations of DLCPA: (a) removes the self-supervised learner (SSL) on the basis of DLCPA; (b) trains the plastic learner based on a randomly initialized temporary classifier. As illustrated in the first row of table 4, the performance degradation of (a) compared with DLCPA proves the necessity of a self-supervised technique for DLCPA. Besides, the second row of Table 4 shows (b) performs lower performance than DLCPA on all settings. The reason for this phenomenon is that the classifier of DLCPA implicitly learns the feature pattern of the stable learner. Therefore the classifier is able to guide the plastic feature learner to extract such features, thus transferring knowledge from the former tasks to the latter and alleviating forgetting. However, (b) lacks this ability and suffers performance degradation.

**Effectiveness of stable learner updating stage.** We next construct two variations by substituting different stable-learner parameters updating strategies, including (c) exponential moving average (EMA, the update factor is set to the conventional 0.999 (He et al., 2020)) with plastic-learner parameters and (d) directly copying the parameters of the plastic learner. As shown in Table 4, DLCPA outperforms both (c) and (d), indicating that the proposed cumulative average update is more suitable for incremental learning.

**Effectiveness of classifier training stage.** The last variation (e) is constructed by training the classifier with the features extracted by the plastic learner. And the fifth row of Table 4 shows a poor performance of (e), because of the mismatching between the classifier and the stable learner.

## 5.4 SENSITIVE ANALYSIS OF $\lambda$

This section analyzes the sensitivity of the balance parameter $\lambda$ (weighting the supervised loss) in equation 6. We select six different values of $\lambda$ and rerun the experiments on 10-split CIFAR-100 and 10-split Tiny-ImageNet, including 0, 0.1, 1, 10, 100, and 1000. Figure 3 illustrates the results, indicating how the accuracy of DLCPA changes as $\lambda$ increases. Interestingly, we observed a similar trend on both datasets. When $\lambda = 0$, DLCPA performs worse under both Task-IL and Class-IL settings, as the supervised loss does not contribute to the plastic learner training, and DLCPA has poor discriminability for each task. As $\lambda$ increases, the accuracy grows and reaches the peak when $\lambda = 10$. In conclusion, the performance DLCPA is sensitive to a small value of $\lambda$, but remains stable for larger weights like 10 for both CIFAR-100 and Tiny-ImageNet.

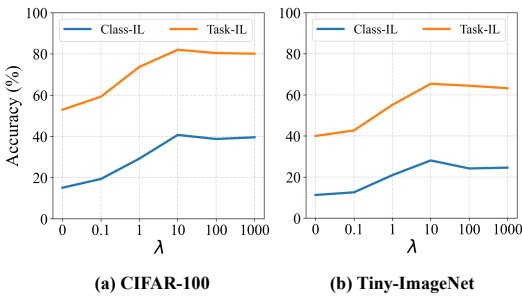

**(a) CIFAR-100**   **(b) Tiny-ImageNet**

Figure 3: Incremental learning results (ACC %) of DLCPA with MoCoV2 under various $\lambda$ (weighting the supervised loss, see Eq. 6) on CIFAR-100 (a) and Tiny-ImageNet (b).

## 6 CONCLUSION

Through the exploratory experiments on STL, we discover that the knowledge of distinct extractors can be integrated by parameter averaging. This insight leads to the proposal of A Dual-Learner framework with Cumulative Parameter Averaging (DLCPA) for exemplar-free IL. DLCPA employs a dual-learner design, with a plastic learner for new-task representations and a stable learner for accumulating all knowledge learned by the plastic learner. Additionally, task-specific classifiers are alternately updated to align with the stable learner. Experiments on CIFAR-100 and Tiny-ImageNet demonstrate that DLCPA achieves state-of-the-art performance on both exemplar-free IL.

Nevertheless, DLCPA has certain limitations, such as the reliance on clear task boundaries for new knowledge organization and the assumption of task equivalence. Future work will delve into knowledge organization techniques and broaden the application scenarios of DLCPA.

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

## A  PSEUDO CODE OF DLCPA TRAINING

As depicted in Figure 2, the training of the DLCPA framework for task $t$ involves a three-stage loop: plastic learner learning, stable learner updating, and classifier training. Within each task, the three-step iteration **repeats** multiple times until the model converges. We outline the training process of DLCPA in Algorithm 1.

---

**Algorithm 1:** Pseudo code of DLCPA Training

**Input:** Datasets $\{D_1, \ldots, D_T\}$
**Output:** Stable learner $\Phi$; Classifiers $\{\Gamma_1, \ldots, \Gamma_T\}$
1  Initialize $\Theta$, $\Phi$, $\{\Gamma_1, \ldots, \Gamma_T\}$;
2  **for** $t = 1, \ldots, T$ **do**
3      **for** $(\mathcal{X}, \mathcal{Y}) \sim D_t$ **do**
4          $\Theta \leftarrow$ Plastic_Learner_Learning$(\mathcal{X}, \mathcal{Y}, \Theta, \Gamma_t)$ // Refer to equation 6
5          $\hat{\Phi} \longleftarrow$ Stable_Learner_Updating$(\Theta, \Phi)$ // Refer to equation 8
6          $\Gamma_t \longleftarrow$ Classifier_Training$(\mathcal{X}, \mathcal{Y}, \hat{\Phi}, \Gamma_t)$ // Refer to equation 9
7      $\Phi \longleftarrow \hat{\Phi}$;

---

## B  SINGLE TASK LEARNING

Single Task Learning (STL) achieves an upper-bound performance in Task-IL (Yoon et al., 2020). It trains a distinct network for each task and makes inferences based on the task identifications of queries. This study explores a more stable version of STL that initializes the new-task model $\Theta_t$ with the previous model $\Theta_{t-1}$, as detailed in Algorithm 2.

## C  EFFECTIVENESS ANALYSIS OF STL-ME

This section analyzes the effectiveness of STL-me, which is built by averaging the feature extractors of STL in the parameter space.

---

**Algorithm 2:** Pseudo code of Single Task Learning

---

**Input:** Datasets $\{D_1, \ldots, D_T\}$
**Output:** Feature extractors $\Theta\_arr = \{\Theta_1, \ldots, \Theta_T\}$;
Classifiers $\Gamma\_arr = \{\Gamma_1, \ldots, \Gamma_T\}$
1 Initialize $\Theta$, $\Theta\_arr = \{\ \}$, $\Gamma\_arr = \{\ \}$;
2 **for** $t = 1, \ldots, T$ **do**
3     Initialize $\Gamma_t$
4     **for** $(\mathcal{X}, \mathcal{Y})$ *in* $D_t$ **do**
5         $\Theta \leftarrow L_{sl}(\Gamma_t^\top f(\mathcal{X}; \Theta), \mathcal{Y})$
6     $\Theta_t = \Theta$
7     $\Theta\_arr = \Theta\_arr \cup \{\Theta_t\}$
8     $\Gamma\_arr = \Gamma\_arr \cup \{\Gamma_t\}$

---

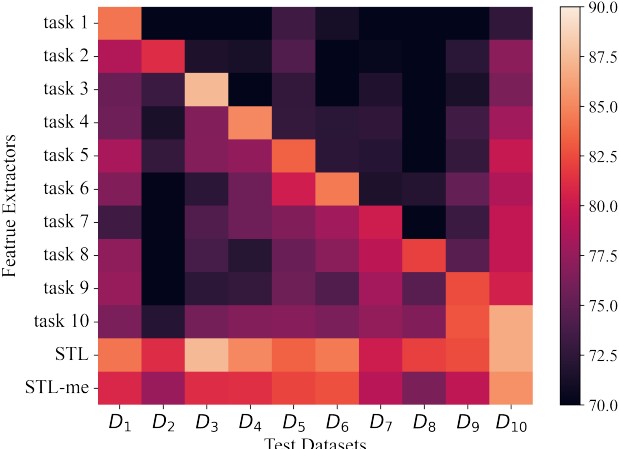

Figure 4: Effectiveness evaluation of various feature extractors on 10-split CIFAR100. The heatmap presents the linear probe accuracy (%) on the dataset of each task by using various feature extractors. These include individual task feature extractors, STL (i.e., choosing the best extractor for every single task), and STL-me (using the mean extractor).

### C.1 DISCRIMINABILITY OF MEAN EXTRACTOR

We first empirically evaluate the discriminability of the features extracted by STL-me through linear probing on 10-split CIFAR100. Figure 4 presents the results for each task-specific extractor in STL, STL, and STL-me. It is evident that each task-specific extractor can only extract discriminative features for its corresponding task data, failing for other tasks. In contrast, STL-me is capable of extracting informative features and nearly matches the STL performance for almost all tasks. This suggests that the parameter averaging operation effectively consolidates knowledge across all extractors.

### C.2 PERFORMANCE REDUCTION BOUND OF STL-ME

Next, we provide an upper bound for the performance reduction of STL-me compared to STL. Prior to the analysis, we introduce two assumptions.

Consider a well-trained STL model with $T$ task feature extractors $\{\Theta_1, \ldots, \Theta_T\}$ and classifiers $\{\Gamma_1, \ldots, \Gamma_T\}$. The loss function $L_{sl}(\Gamma_t^\top f(x_t; \Theta_t), y_t)$ is Lipschitz continuous (Bonicelli et al., 2022) with respect to $\Theta_t$. That is, for any task $t$, there exists a positive real number $k$ such that

$$(L_{sl}(\Gamma_t^\top f(x_t; \Theta_t + \Delta\Theta), y_t) - L_{sl}(\Gamma_t^\top f(x_t; \Theta_t), y_t))^2$$
$$\leq k||\Delta\Theta||^2. \tag{11}$$

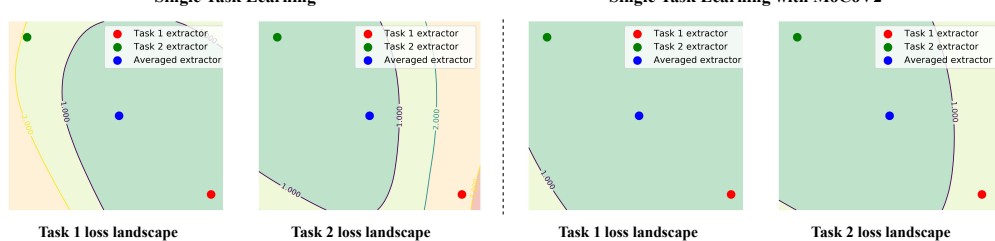

Figure 5: Cross-entropy validation loss surface for task 1 and task 2 on CIFAR-100. STL is updated with supervised loss (left) in comparison to with both supervised and self-supervised loss (right).

For all $T$ tasks, the number of iterations is bounded by an integer $S$.

Assuming that Assumptions C.2 and C.2 hold, let $\eta$ be the learning rate. Then, the loss of task $t$ after parameter averaging is upper bounded by

$$
(L_{sl}(\Gamma_t^\top f(x_t; \frac{1}{T}\sum_{i=1}^{T}\Theta_i), y_t) - L_{sl}(\Gamma_t^\top f(x_t; \Theta_t), y_t))^2
$$
$$
\leq k(\frac{TS\eta}{2})^2. \tag{12}
$$

*Proof.* Let $\Theta_{t,s}$ denote the feature extractor that is further updated $s$ steps after the completion of learning task $t$. Specifically, $\Theta_t = \Theta_{t,0}$ represents the extractor specific to task $t$. Then we have

$$
L_{sl}(\Gamma_t^\top f(x_t; \frac{1}{T}\sum_{i=1}^{T}\Theta_i), y_t)
$$
$$
\leq L_{sl}(\Gamma_t^\top f(x_t; \frac{1}{T}(\Theta_{t,-(t-1)S} + \cdots + \Theta_{t,0} + \ldots
$$
$$
+ \Theta_{t,(T-t)S}), y_t) \tag{13}
$$
$$
\leq L_{sl}(\Gamma_t^\top f(x_t; \frac{1}{T}(T\Theta_t + \frac{T^2}{2}S\eta e), y_t)
$$
$$
= L_{sl}(\Gamma_t^\top f(x_t; \Theta_t + \frac{T}{2}S\eta e), y_t),
$$

where $e$ is a unit vector in the loss ascent direction.

Therefore, by using the Lipschitz continuity, we derive

$$
(L_{sl}(\Gamma_t^\top f(x_t; \frac{1}{T}\sum_{i=1}^{T}\Theta_i), y_t) - L_{sl}(\Gamma_t^\top f(x_t; \Theta_t), y_t))^2
$$
$$
\leq k(\frac{TS\eta}{2})^2. \tag{14}
$$

$\square$

Theorem C.2 suggests that the flatness of the loss landscape, denoted as $k$ in Eq. equation 14, is pivotal to the performance of STL-me. To explore the sharpness, we illustrate the loss landscape of STL on the first two tasks of CIFAR100 in Figure 5, following the approach of Mirzadeh et al. (2021). As observed, the loss surfaces of both tasks 1 and 2, over the linear combination of the task-1 feature extractor and task-2 feature extractor, exhibit smoothness. This observation leads us to conclude that Assumption C.2 holds with a lower $k$ value in the range between the averaged extractor and task-specific extractors. Consequently, their average provides a suitable solution for both tasks. This is plausible as different incremental tasks share similar underlying knowledge, even if the classes to be recognized are distinct, a common assumption in incremental learning (Ke et al., 2020).

Table 5: Hyperparameters for the baseline models and our DLCPA. In this context, "KD" stands for "Knowledge Distillation"

| Methods | Hyperparameters |
|---------|-----------------|
| GEM (Lopez-Paz & Ranzato, 2017) | learning rate: 0.03 (CIFAR-100), 0.05 (Tiny-ImageNet) 
 batch size: 32 (CIFAR-100), 64 (Tiny-ImageNet) |
| DER (Buzzega et al., 2020) | learning rate: 0.03 batch size: 32 KD weight: 0.1 |
| DER++ (Buzzega et al., 2020) | learning rate: 0.03 batch size: 32 KD weight $\alpha$: 0.1 $\beta$: 0.5 |
| OWM (Zeng et al., 2019) | learning rate: 0.1 batch size: 64 |
| ILCOC (Sun et al., 2021) | learning rate: 2e-3 batch size: 32 $\alpha_1$: 0.7 $\alpha_2$: 0.5 |
| PASS (Zhu et al., 2021) | learning rate: 1e-4 (Tiny-ImageNet), 5e-4 (CIFAR-100) 
 batch size: 64 (CIFAR-100), 32 (Tiny-ImageNet) 
 KD weight: 10 (Tiny-ImageNet), 0.2 (CIFAR-100) 
 prototype weight: 0.05 (CIFAR-100), 0.5 (Tiny-ImageNet) |
| DCPOC (Sun et al., 2023a) | learning rate: 5e-5 
 batch size: 8 (CIFAR-100), 32 (Tiny-ImageNet) 
 $\lambda_1$: 20 (CIFAR-100), 10 (Tiny-ImageNet) 
 $\lambda_2$: 10 (CIFAR-100), 0.01 (Tiny-ImageNet) |
| FAS (Miao et al., 2022) | learning rate: 0.001 (CIFAR-100), 5e-4 (Tiny-ImageNet) 
 batch size: 32 (CIFAR-100), 16 (Tiny-ImageNet) |
| DLCPA (ours) | learning rate: 0.005 (CIFAR-100), 0.02 (Tiny-ImageNet) 
 batch size: 32 (CIFAR-100), 512 (Tiny-ImageNet) 
 self-supervised loss weight $\lambda$: 10 |

Table 6: Dataset statistics.

| Dataset | CIFAR-100 | Tiny-ImageNet |
|---------|-----------|---------------|
| Input size | $3 \times 32 \times 32$ | $3 \times 64 \times 64$ |
| # Classes | 100 | 200 |
| # Training samples per class | 450 | 450 |
| # Validation samples per class | 50 | 50 |
| # Testing samples per class | 100 | 100 |

On the other hand, reducing $k$ diminishes the upper bound of performance degradation and further enhances the effectiveness of parameter averaging. This insight motivates us to incorporate self-supervised techniques during learning. As shown in Figure 5 (right), training with self-supervised learners flattens the loss landscapes. Therefore, we equip DLCPA with self-supervised learners to prompt the plastic learner to concentrate more on task-independent knowledge, thereby enhancing the impact of cumulative average updating.

## D EXPERIMENTAL DETAILS

This section provides some details of our experiments, including the experimental environment, the hyperparameters used for the proposed DLCPA and main baselines under comparison, and the statistics of datasets.

### D.1 EXPERIMENTAL ENVIRONMENT

All experiments reported in our manuscript and appendix are conducted on a workstation running OS Ubuntu 16.04 with 18 Intel Xeon 2.60GHz CPUs, 256 GB memory, and 6 NVIDIA RTX3090 GPUs. All methods are implemented with Python 3.8.

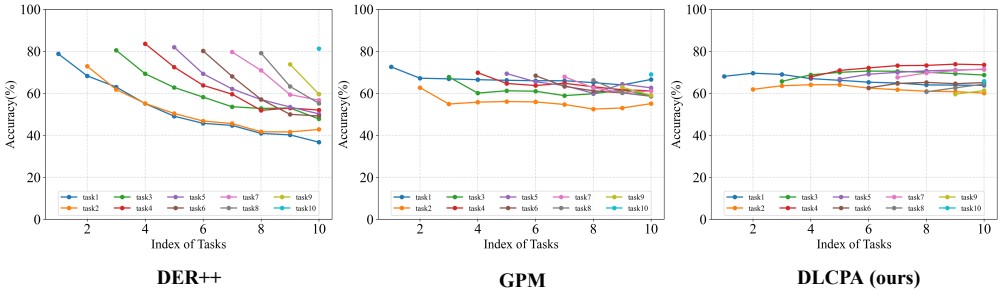

Figure 6: Diagram of ACC (%) for each task during task incremental learning on Tiny-ImageNet of DER++ (left), GPM (middle), and DLCPA (right).

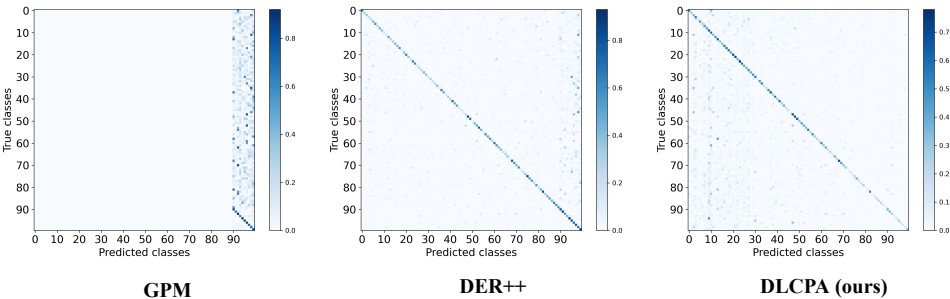

Figure 7: Confusion matrix of DER++, GPM, and DLCPA on 10-Split CIFAR-100.

## D.2 LIST OF HYPERPARAMETERS

Table 5 summarizes the hyperparameters that were tuned for the baseline methods we implemented, as well as for our proposed DLCPA.

## D.3 DATASET STATISTICS

In this work, we utilize two classification datasets, CIFAR-100 (Krizhevsky, 2012) and Tiny-ImageNet (Stanford, 2015), to evaluate the proposed DLCPA. The statistical details of these two datasets are provided in Table 6.

## E ADDITIONAL COMPARISONS

This section presents additional experimental results. These supplementary experiments encompass (i) an analysis of plasticity and stability, (ii) a comparison of confusion matrices, (iii) a comparison of Class-IL with varying numbers of tasks, (iv) a comparison with mask-based incremental learning methods, (v) a comparison with a larger network backbone, and (vi) an analysis of DLCPA's robustness to task order.

## E.1 PLASTICITY AND STABILITY ANALYSIS

We plot the performance of each task during the incremental learning of DER++, GPM, and DLCPA to examine their plasticity and stability. As depicted in Figure 6, for DER++, peak performance for each task is achieved after training completion, and the maximum performance for each task is the highest among the three methods. This observation attests to DER++'s high plasticity. However, its stability is lacking, as each task's performance declines immediately after learning subsequent tasks. GPM, on the other hand, maintains a more stable performance for each task, suggesting that its gradient constraint mechanism ensures high stability. However, GPM's gradient projection technique

may limit its ability to learn new knowledge, resulting in insufficient plasticity. DLCPA exhibits both high plasticity and stability. Furthermore, the performance of most tasks (except for task 1 and task 2) under DLCPA gradually improves during subsequent task learning due to its alternating update design and self-supervised techniques, which enhance its ability to transfer knowledge to old tasks.

### E.2 CONFUSION MATRIX COMPARISION

Classification bias is a well-known challenge in Class-IL (Wu et al., 2019). We compute the confusion matrices of DER++, GPM, and DLCPA to assess the balance of their classifications. As illustrated in Figure 7, GPM's classification leans towards the classes of the last task. This is expected, as GPM is a Task-IL method and lacks a specific design to counteract classification bias. For DER++, the phenomenon of classification bias is significantly reduced, thanks to the stored old-task exemplars. Notably, DLCPA, even under an exemplar-free setting, achieves a task-wise balance similar to that of the exemplar-memory based DER++.

Table 7: Class incremental learning results (ACC %) on CIFAR-100 for various task numbers (5, 10, 20). Results marked with "†" are derived from Smith et al. (2021).

| Task Number | 5 tasks | 10 tasks | 20 tasks |
|---|---|---|---|
| DGR† | 14.4 | 8.1 | 4.1 |
| LwF† | 17.0 | 9.2 | 4.7 |
| DI† | 18.8 | 10.9 | 5.7 |
| ABD† | 43.9 | 33.7 | 20.0 |
| PASS | 45.2 | 30.8 | 17.4 |
| DCPOC | 33.1 | 27.5 | 20.5 |
| DLCPA w/ MoCoV2 | **46.3** | **40.1** | **31.0** |

### E.3 CLASS-IL RESULTS WITH VARIOUS NUMBER OF TASKS

We further evaluate the performance of DLCPA across different task sequence lengths under Class-IL. Following Smith et al. (2021), we conduct experiments on CIFAR-100 with 5, 10, and 20 tasks. The methods compared include DGR (Shin et al., 2017), LWF (Li & Hoiem, 2017), DI (Yin et al., 2020), ABD (Smith et al., 2021), PASS (Zhu et al., 2021), and DCPOC (Sun et al., 2023a), with their performance detailed in Table 7. In the 5 task setting, DLCPA marginally surpasses the second-best method, PASS. However, when faced with a longer sequence of 20 tasks, DLCPA demonstrates a more significant performance gap, outperforming the second-best by 10.5%. These results validate the superiority of DLCPA when learning longer tasks.

Table 8: Comparison (ACC %) with Mask-based methods on a 10-split Tiny-ImageNet. Results marked with (†) are derived from Wang et al. (2022).

| Methods | Tiny-ImageNet | |
|---|---|---|
| | Class-IL | Task-IL |
| PackNet† (Mallya & Lazebnik, 2018) | - | 61.88±1.0 |
| LPS† (Wang et al., 2020) | - | 63.37±0.8 |
| SparCL† (Wang et al., 2022) | 20.75±0.9 | 52.19±0.4 |
| DLCPA w/ MoCoV2 (Ours) | **29.21±0.3** | **65.90±0.4** |

### E.4 COMPARISON WITH MASK-BASED METHODS

In line with Wang et al. (2022), we conducted experiments on Tiny-ImageNet, with the results presented in Table 8. The mask-based methods compared include PackNet (Mallya & Lazebnik, 2018), LPS (Wang et al., 2020), and SparCL (Wang et al., 2022). Notably, SparCL is an exemplar-based method and maintains an additional exemplar memory with 500 samples. As observed, DLCPA sig-

nificantly outperforms all comparison methods. Furthermore, DLCPA's performance is more stable, exhibiting the lowest variance.

Table 9: Result (ACC %) using ResNet34 as the network backbone on a 10-split CIFAR-100 and a 10-split Tiny-ImageNet.

| Methods | CIFAR-100 | |
| --- | --- | --- |
| | Class-IL | Task-IL |
| DER++ (Buzzega et al., 2020) | 16.98 | 51.34 |
| GPM (Saha et al., 2021) | - | 66.16 |
| PASS (Zhu et al., 2021) | 28.62 | 75.69 |
| DLCPA (ours) | **40.69** | **82.01** |
| Methods | Tiny-ImageNet | |
| | Class-IL | Task-IL |
| DER++ (Buzzega et al., 2020) | 19.54 | 50.48 |
| GPM (Saha et al., 2021) | - | 60.28 |
| PASS (Zhu et al., 2021) | 28.48 | 63.02 |
| DLCPA (ours) | **28.96** | **65.87** |

### E.5   RESULTS WITH LARGER NETWORK BACKBONE

We conducted experiments using ResNet34(He et al., 2016) as the network backbone, with the results reported in Table 9. As observed, DLCPA achieves superior performance compared to three state-of-the-art methods (DER++ (Buzzega et al., 2020), GPM (Saha et al., 2021), and PASS (Zhu et al., 2021)) across all settings, demonstrating DLCPA's robustness to network scale.

Table 10: Incremental learning result (ACC %) of DLCPA with MoCoV2 on a 10-split CIFAR-100 with five random task orders.

| Methods | CIFAR-100 | |
| --- | --- | --- |
| | Class-IL | Task-IL |
| Task order 1 | 40.29 | 82.04 |
| Task order 2 | 38.76 | 81.39 |
| Task order 3 | 39.63 | 81.66 |
| Task order 4 | 35.52 | 81.23 |
| Task order 5 | 40.23 | 81.31 |
| Average | 38.89±1.77 | 81.53±0.29 |

### E.6   RESULTS WITH RANDOM TASK ORDERS

This subsection investigates the robustness of DLCPA to task order. To this end, we randomly shuffle the task order on CIFAR-100 and retrain DLCPA incrementally five times. Table 10 presents the results for each order and the average performance. As observed, except for an obvious performance decline in Order 4 of Class-IL, DLCPA exhibits relatively stable performance across different scenarios, indicating the robustness of DLCPA to the task order.

