# OpenReview forum: "Towards Plastic and Stable Exemplar-Free Incremental Learning: A Dual-Learner Framework with Cumulative Parameter Averaging"
_ICLR.cc/2024/Conference — ICLR 2024 Conference Withdrawn Submission_

### Official Review · Reviewer_Gkrh · 2023-10-22

**Soundness:** 2 fair
**Presentation:** 3 good
**Contribution:** 2 fair
**Rating:** 5
**Confidence:** 3

**Summary:**

This paper proposes a method for incremental learning for vision-based classification tasks. The method learns a feature extractor with two components: a stable learner and a plastic learner. A new plastic learner is trained for each task. The stable learner averages the weights of all of the previous plastic learners. In addition to the two feature extractors, a separate classifier is trained for each task. Therefore, the architecture achieves a memory growth that is linear in the number of tasks, but with a much smaller coefficient than storing the data or all of the feature extractors. A crucial part of the proposed method is to use a self-supervised loss for training the plastic learner.

**Strengths:**

## Presentation
- The paper is clearly written for the most part.

## Contribution
- The core idea behind the presented method is simple.
- The method seems to work well on Tiny-imagenet
- The analysis of how the proposed stable-plastic architecture can help is nice and simple.

**Weaknesses:**

## Presentation
- The "Analysis" section contains the problem definition and explanation of the baselines, which are background information. It would be better to split this information into a clear background section.
- The task settings "Task-IL" and "Class-IL" are never defined in the paper.
- The paper assumes that storing previous models or datasets is so costly that it is a big disadvantage to the memory-based methods. However, from a lay perspective, it would seem that storage is very cheap. I suggest the authors include more motivating discussion about why the memory-based IL methods have a disadvantage.

## Contribution
- The self-supervised learning (SSL) objective is introduced without appropriate analysis and baselines.
    - From the ablation studies, we see that the SSL objective is a crucial component of the proposed method. However, I don't see how SSL is specific to the proposed method. Without applying SSL objectives to at least some of the most competitive baselines, we don't know how much of the improvement from the proposed method is actually due to the stable-plastic learner architecture.
    - Three different SSL losses are proposed to be used with the method. The best results in each of the datasets are split between each of the SSL objectives. On some datasets, choosing a different SSL objective would lead to the proposed method no longer beating the baselines. This seems like a weakness of the method, in that you have to train three versions of it in order to pick which one you want to deploy.

**Questions:**

- Which SSL objective should a user choose?
- To which baselines is the SSL objective applicable?
    - How well would those baselines do with the SSL objective?
- Is there a practical task, where one of the methods considered in this paper would work well, but storing all of the data would become prohibitively expensive?

---

### Official Review · Reviewer_CE6t · 2023-10-31

**Soundness:** 2 fair
**Presentation:** 2 fair
**Contribution:** 2 fair
**Rating:** 3
**Confidence:** 3

**Summary:**

This paper argues, and attempts to empirically demonstrate, that cumulatively averaging the parameters of models trained on individual tasks is a promising approach for continual learning. First, using task-incremental versions of Split MNIST and Split CIFAR-100, the paper shows that by averaging the parameters of task-specific models trained on each task individually, only a moderate amount of performance is lost relative to using the single-task models directly. Motivated by this, the paper then combines cumulative parameter averaging with self-supervised learning to engineer a multi-module approach for continual learning with strong empirical performance on both task- and class-incremental benchmarks.

**Strengths:**

I think it is an intriguing claim of the paper that a promising approach for continual learning is to cumulatively average the parameters of models trained on individual tasks. As the paper discusses, there is some other recent work (e.g., CLS-ER and DualNet) that builds on this approach, but I think that a principled demonstration of the benefits of the parameter averaging approach is still missing.

I think it is also a strength that the authors demonstrate that their proposed approach can work with different self-supervised learning techniques.

**Weaknesses:**

Unfortunately, I do not think that this paper provides a clear or principled demonstration of the benefit of parameter averaging.

Regarding the experiments of Table 1, I am not convinced that these results are indicative that (cumulative) parameter averaging is a promising method. On Split MNIST, STL-me drops to less than 97%, and even after fine-tuning the classifier the performance is still below 99%. This is on two-way classification tasks of MNIST digits. What would be the performance of a linear classifier in this situation? That is, what would be obtained if you do “classifier fine-tunining” directly on the raw pixels? Another baseline that I think would be good to include in Table 1 is a “small version of STL”, in which there is a separate model for each task (i.e., like in STL), but each task-specific model is relatively small, such that the total parameter count of the combined models is the same as STL-me. Would STL-me outperform this baseline?

Perhaps the authors would argue that the strong performance of their engineered multi-module method in Tables 2 and 3 demonstrates the effectiveness of cumulative parameter averaging for continual learning, but I do not think that is the case. For example, an aspect present in DLCPA but not in the compared baselines (at least not in most of them) is self-supervised learning. Indeed, from Table 4 it seems that if the self-supervised learning aspect is left out, the superior performance of DLPCA disappears.

An intriguing property of DLCPA is that it is able to distinguish between classes from different tasks (e.g. Appendix E2). This is surprising to me, at least at first, because my understanding is that DLCPA is actually never trained to distinguish between classes from different tasks. Could the authors attempt to explain why it is nevertheless able to do so?

Although code is provided, I could not find instructions on how to run or reproduce the experiments reported in the paper.

**Questions:**

I would like to encourage the authors to include a clear / principled demonstration of the benefit of cumulative parameter averaging for continual learning.

I would be happy to actively engage in the discussion period.

---

### Official Review · Reviewer_ikP1 · 2023-10-31

**Soundness:** 2 fair
**Presentation:** 3 good
**Contribution:** 3 good
**Rating:** 5
**Confidence:** 4

**Summary:**

Incremental learning (IL) refers to the ability to continuously learn new knowledge from a series of tasks. The main challenge in incremental learning is to achieve high plasticity for new task learning without de-stabilizing learnings based on older tasks. In this paper, the authors propose a new framework, Dual-Learner framework with Cumulative Parameter Averaging (DLCPA), to solve exemplar-free incremental learning problems. The example-free IL, in general, requires advanced distilling techniques. The dual-learner uses a plastic learner for acquiring new task knowledge, and a stable learner for accumulating all previously learned knowledge. The experimental results show that DLCPA outperforms several state-of-the-art baselines under both exemplar-free TaskIL and Class-IL settings.

**Strengths:**

Originality & Significance. Dual-learning architectures have previously been applied in exemplar-based IL methods. These prior methods depend on old-task exemplars to maintain the stable model and prevent forgetting. In contrast, DLCPA employs a cumulative average update strategy for the stable learner. The foundation of the approach is that averaging in the parameter space of single-task learner could potentially be an effective strategy for preserving knowledge across all tasks. This is not trivial.

Quality & Clarity. The paper is well-written. And the explanations are clear. There are not many grammatical errors.

The authors compare their algorithm to other SOTA algorithms using popular benchmark datasets such as CIFAR and Tiny-ImageNet.

**Weaknesses:**

It is not clear to me how the averaging works if the parameter space is highly nonlinear with several local optimal such that when averaged is not optimal any longer. The authors do not address this issue through theoretical analysis and assumptions under which their method may fail. However, the empirical results are rather convincing. It may be helpful to include tasks where the parameter space has nonlinearities.

**Questions:**

It is not clear to me how the averaging works if the parameter space is highly nonlinear with several local optimal such that when averaged is not optimal any longer. The authors do not address this issue through theoretical analysis and assumptions under which their method may fail.

Have the authors considered having a regularization term that may help and smooth out the models which is more generalizable to nonlinear parameter space?

---

### Official Review · Reviewer_bahq · 2023-11-03

**Soundness:** 2 fair
**Presentation:** 2 fair
**Contribution:** 1 poor
**Rating:** 3
**Confidence:** 2

**Summary:**

The authors present a method for lifelong learning for neural networks in which they perform parameter averaging for a feature extractor and learn a small classifier layer on top for every task. They show that this improves performance on lifelong learning setups in CIFAR and TinyImagenet datasets.

**Strengths:**

1. The paper is well written and easy to follow
2. The intuition behind the work is clear.

**Weaknesses:**

1. **Novelty**: My major concern is with the novelty of the work. I don't believe that it tells the community something that we did not already know. For instance, works like SKILL https://arxiv.org/pdf/2305.15591.pdf, BatchEnsemble: https://arxiv.org/pdf/2002.06715.pdf and the entire field of pretrained representations for transfer tells us that sharing parameters across tasks is useful and one can get appreciable performance with linear probing on frozen representations. Therefore it's not clear to me that this work provides much new insight.
2. **Insufficient Evaluation**: I don't think this work provides sufficient evaluation, performing experiments mainly on small and old datasets. SKILL-102 (https://arxiv.org/pdf/2305.15591.pdf) for instance provides many more challenging incremental tasks useful for lifelong learning evaluation. The evaluation can be significantly improved.

**Questions:**

I don't have any additional questions.